# Antibodies-Abzymes with Antioxidant Activities in Two Th and 2D2 Experimental Autoimmune Encephalomyelitis Mice during the Development of EAE Pathology

**DOI:** 10.3390/molecules27217527

**Published:** 2022-11-03

**Authors:** Anna S. Tolmacheva, Kseniya S. Aulova, Andrey E. Urusov, Vasiliy B. Doronin, Georgy A. Nevinsky

**Affiliations:** Institute of Chemical Biology and Fundamental Medicine, Siberian Division of the Russian Academy of Sciences, 630090 Novosibirsk, Russia

**Keywords:** EAE model, Th and 2D2 mice, catalytic antibodies, peroxidase and oxidoreductase activities

## Abstract

The exact mechanisms of multiple sclerosis development are still unknown. However, the development of EAE (experimental autoimmune encephalomyelitis) in Th and 2D2 mice is associated with the infringement of the differentiation profiles of bone marrow hematopoietic stem cells which are bound with the production of compounds that are harmful for human autoantibodies-abzymes that hydrolyze myelin oligodendrocyte glycoprotein, myelin basic protein, and DNA. It showed that autoimmune patients’ antioxidant IgG antibodies oxidise some compounds due to their peroxidase (H_2_O_2_-dependent) and oxidoreductase (H_2_O_2_-independent) activities more effectively than those in healthy humans can. It was interesting to identify whether the redox activities of the antibodies change during the development of autoimmune diseases. Here, we analyzed the change in these redox activities of the IgGs from the blood of Th and 2D2 mice, which corresponded to different stages of the EAE development. The peroxidase activity in the oxidation of ABTS (2,2′-azino-bis(3-ethylbenzothiazoline-6-sulfonic acid)) in the Th (4-fold) and 2D2 (2-fold) mice IgGs, on average, is higher than the oxidoreductase activity is. The peroxidase activity of the Th (1.9-fold) and 2D2 (3.5-fold) mice IgGs remarkably increased during the 40 days of the spontaneous development of EAE. Forty days after the immunization of the MOG peroxidase activity, the IgGs of the Th and 2D2 mice increased 5.6–6.0 times when they were compared with those that presented no increase (3 months of age). The mice IgGs were oxidized with 3,3′-diaminobenzidine (2.4–4.3-fold) and o-phenylenediamine (139–143-fold) less efficiently than they were with ABTS. However, the temper of the change in the IgG activity in the oxidation of these substrates during the spontaneous and MOG-induced development of EAE was close to that which occurred for ABTS. All of the data show that the IgG peroxidase and oxidoreductase activities of EAE mice can play an important role in their protection from toxic compounds and oxidative stress.

## 1. Introduction

Multiple sclerosis (MS) is a chronic autoimmune-associated demyelinating disease of the central nervous system (CNS). MS’ etiology still yet remains unclear, and the most accepted theory of multiple sclerosis pathogenesis is associated with a leading role in the destruction of myelin due to the inflammation related to autoimmune reactions [1]. Autoimmune diseases (AIDs) were proposed first to possibly be originated from defects of the hematopoietic stem cells (HSCs) [2,3,4,5]. The spontaneous and antigen-induced evolutions in MRL-lpr/lpr mice that were prone to systemic lupus erythematosus (SLE) [3,4,5] and EAE-prone C57BL/6 [6,7,8], Th [9,10], and 2D2 mice [11] were later demonstrated to be the consequence of the immune system-specific readjustment of bone marrow HSCs. The immune system defects in AIDs which are connected with specific changes in the profiles of bone marrow HSC differentiation are associated with the production of particular catalytic auto-antibodies-abzymes hydrolyzing many different compounds, including polysaccharides, DNAs, RNAs, peptides, and proteins [3,4,5,6,7,8,9,10,11,12,13,14,15,16,17].

Abs-abzymes, against the transition states of many different chemical reactions, catalyzing more than 170 various reactions, are new biologic catalysts (for review, see [12,18,19,20]). Natural catalytic Abzs-destroying oligopeptides, proteins, polysaccharides, DNAs, and RNAs were revealed in the blood sera of patients with several AIDs (for review, see [12,13,14,15,16,17]). With several exceptions, the Abs-abzymes demonstrating these catalytic activities in healthy mammals are absent or usually possess extremely low activity levels. The auto-antibodies having enzymatic activities (DNase, RNase, protease, amylase, and ATPase) were revealed to be statistically significant and the earliest markers of the development of many AIDs [12,13,14,15,16,17].

The DNA-hydrolyzing abzymes of MS patients and SLE are dangerous since they are cytotoxic; they penetrate through the cellular and nuclear membranes and they stimulate cell apoptosis which plays a very negative role in SLE and MS pathogenesis [21,22]. The abzymes against myelin oligodendrocyte glycoprotein (MOG) and myelin basic protein (MBP) with protease activities in SLE and MS patients may attack and hydrolyze the MBP of the myelin-proteolipid sheath of the axons. Consequently, these abzymes may also play a very negative role in MS and SLE pathogenesis [13,14,15,16,17].

Several different oxygen species in various organisms (O_2_·−, H_2_O_2_, and OH·) are intermediates of aerobic respiration; they may also appear through exposure to ionizing radiation. These oxygen species are potent oxidants, attacking and destroying various cellular lipids, DNA, proteins, etc. [23,24,25]. Oxidative damages to different cells compounds are considered to be essential factors in ageing due to mutagenesis and carcinogenesis. Several antioxidant canonical enzymes (superoxide dismutases, catalases, and glutathione peroxidases) procure the critical mechanisms of protection from oxidative stress for blood and cell components [23,24,25,26,27,28,29]. The antioxidant enzymes of humans and mammals are present mostly in different cells, but only the poor redox enzymatic activities can be detected in the blood because they relatively quickly lose activities in the blood [28,29]. In contrast to the canonical antioxidant enzymes, various antibodies exist in the blood for a long time (1–3 months) [30]. Therefore, it was intriguing to identify whether blood catalytic Abzs could participate in the protection of humans and mammals from oxidative stress. It was shown that the IgG antibodies of Wistar rats in the presence (peroxidase activity) and the absence of hydrogen peroxide (peroxide-independent oxidoreductase activity), in a similar way to the canonical peroxidase of horseradish, could oxidize 3,3′-diaminobenzidine (DAB) and other different compounds well [31,32,33,34,35]. The Wistar rat’s IgGs also showed superoxide dismutase and catalase activities [36].

Undeniable evidence that both peroxidase and oxidoreductase catalytic activities are intrinsic to the IgG levels of healthy mammals has been demonstrated using different methods [31,32,33,34,35,36,37,38,39]; for an overview, see [40]. The redox activities of Abs from healthy volunteers and patients with MS and SLE were compared [39]. The peroxidase and oxidoreductase average activities of the IgG antibodies of patients with SLE were statistically significantly higher in comparison with the Abzs from the healthy volunteers. The MS patients’ IgG levels demonstrated that these peroxidase and oxidoreductase activities are higher than the healthy donors’ are, but lower than the SLE abzymes are [39]. These data showed that abzymes could serve as an additional blood factor of reactive oxygen species detoxification.

Several experimental autoimmune encephalomyelitis (EAE) models mimicking a specific facet of human MS exist, including C57BL/6 [41,42,43], Th [44], and 2D2 [45,46,47] mice. The C57BL/6 mice are characterized by specific T and B lymphocyte’s answers to specific antigens [41,42,43]. The Th model of mice was obtained by crossing transgenic mice that had a particular myelin-specific T-cell receptor with specific myelin-specific Abs heavy chain knock-in mice [44]. The 2D2 TCR (TCRMOG) MS model corresponds to transgenic mice expressing a MOG-specific T cell receptor [45,46,47]. All of these EAE models are characterized by a spontaneous progressive chronic course of EAE and an acceleration of the pathology development after the immunization of mice with MOG [41,42,43,44,45,46,47].

The C57BL/6 [6,7,8], Th [9,10], and 2D2 [11] mice were used to analyze the possible mechanisms over time in spontaneous (untreated mice) and MOG-accelerated EAE development. Some typical medical indicators of EAE development (optic neuritis and other clinical or histological evidence) usually appear relatively late due to spontaneous EAE or faster (0.7–1.5 years) after mice have been immunized with MOG. However, the early stages of EAE development in these mice are observed as early as a few months after their birth.

Three models of EAE mice seemed convenient for elucidating at which stages of developing the autoimmune diseases that the activation of the production of Abzs with redox activities can occur. First, we used C57BL/6 mice to analyze the changes in the redox activities of the IgGs during the onset, acute phase, and remission of EAE pathology [48]. Here, we examined the changes in the activity of antioxidant antibodies during the development of EAE in Th and 2D2 mice, and we compared these processes in three lines of autoimmune mice.

## 2. Results

### 2.1. Experimental Groups of Mice

As mentioned above, the development of EAE in the C57BL/6, Th, and 2D2 mice leads to a specific reorganization of their immune systems consisting of significant changes in the differentiation profiles of the mice HSCs and the increase in different organs’ lymphocytes proliferation [6,7,8,9,10,11]. All of these immune system changes and defects increase the proteinuria and the generation of the catalytically active Abs which split DNA, MBP, and MOG [6,7,8,9,10,11]. In addition, the development of autoimmune diseases in humans leads to an increase in the concentration of antibodies-abzymes with antioxidant redox catalytic activities [37,38,39,40]. It showed recently that the development of EAE in autoimmune-prone C57BL/6 mice was associated with an increase in the redox activities of IgGs [40,48]. Those that are prone to the spontaneous development of EAE in Th and 2D2, such as the C57BL/6 mice, demonstrate during the development of the pathology the changes in the profiles of HSCs differentiation and the production of abzymes that hydrolyze DNA, MBP, and MOG [6,7,8,9,10,11]. To demonstrate the processes of the dysregulation of the immune status in the C57BL/6, Th, and 2D2 mice, the Appendix A indicate that over time, there were changes in a number of mice bone marrow BFU-E, CFU-E, CFU-GM, and CFU-GEMM colony-forming units (Appendix A) [6,7,8,9,10,11]. These changes in the differentiation of the stem cells were associated with the changes in the relative activity of DNA, MOG, and MBP hydrolysis during the spontaneous and MOG-induced development of EAE disorder (Appendix A).

By the age of 1.0–1.5 years, all three of the lines of mice showed typical EAE medical (optic neuritis and other clinical or histological evidence) and immunological indicators [6,7,8,9,10,11,41,42,43,44,45,46,47]. However, three strains of mice in the early stages of the EAE development showed marked differences in the profiles of the HSCs differentiation changes and the production of abzymes with hydrolytic activities [6,7,8,9,10,11]. It was interesting to see how these differences in the C57BL/6, Th, and 2D2 mice immune system changes (Appendix A) could effect on the IgGs relative activities (RAs) in the oxidation of different substrates. In addition, it seemed attractive to compare the overtime patterns of the changes in the RAs of the IgGs with redox functions with those for Abs hydrolyzing DNA, MOG, and MBP during the mice EAE development that had been analyzed earlier [6,7,8,9,10,11].

### 2.2. Criteria Analysis of Catalytic Activities of Antibodies

As previously shown, the optimal substrates for the IgGs with redox activities from healthy rats [31,32,33,34,35], healthy donors, patients with autoimmune diseases [37,38,39,40], and C57BL/6 mice [40,48] are ABTS, DAB, and OPD. We have applied strict criteria to prove that the redox activities of C57BL/6 mice IgGs are their intrinsic properties and to not occur not due to the co-purifying canonical enzymes [48]. Here, we used the known harsh criteria to show that redox activities are the own properties of IgGs from the blood of Th and 2D2 mice. The IgG_mix_ preparations mixtures of 30 IgGs from Th (Th-IgG_mix_) and 2D2 (2D2-IgG_mix_) were electrophoretically homogeneous (Figure 1, lanes 1–4, m1, and m2).

To exclude the possible hypothetical traces of contaminating enzymes, the preparations of IgG_mix_ were subjected to SDS-PAGE. The catalytic activity was revealed after the incubation of the gel in the reaction mixture containing DAB. Yellow-brown precipitate bands were revealed only in the positions of three intact IgG_mix_ (Figure 1B, lanes m1 and m2). Since SDS dissociates the complexes of all the proteins, it reveals the activity only in the gel region corresponding to the intact IgG_mix_ and the absence of any other bands of proteins, and the activity provides sufficient evidence that all of the IgG_mix_ preparations possess an intrinsic peroxidase activity.

### 2.3. Substrates of IgGs

Figure 2 shows the typical kinetic curves of ABTS, DAB, and OPD oxidation by IgG_mix_ preparations of the Th and 2D2 mice in the presence of hydrogen peroxide (peroxidase activity).

Using the kinetic curves, the ABTS, DAB, and OPD oxidation rates were estimated for the individual Th and 2D2 mice (10 preparations in each group) at fixed optimal concentrations of the three substrates. The apparent *k*_cat_ values (*k*_cat_ = V (M/min)/[IgG] (M)) characterizing the oxidation by each preparation were assessed, and we calculated the average values for 10 preparations. Unfortunately, the amount of IgG preparations for a complete analysis of the H_2_O_2_-independent oxidoreductase activity was not enough. With this in mind, we estimated the apparent values of *k*_cat_ using equimolar mixtures of 10 IgG preparations.

### 2.4. In Time Changes in the Peroxidase Activity during the Development of EAE

To evaluate a possible in time changes in the RAs of the 10 individual Th mice IgG preparations before (spontaneous EAE development) and after the mice immunization procedure with MOG, their relative peroxidase activities in the oxidation of ABTS, DAB, and OPD were determined, and the average values for ten preparations were calculated (Figure 3A–C).

In the spontaneous development of EAE, the oxidation rate was the highest in the case of ABTS, and by 40 days, it was 3.9 and 144 times higher than those for DAB and OPD (*p* < 0.05), respectively. After the immunization of the Th mice with MOG, the maximum increase in the peroxidase activity of the IgGs in the oxidation of the three substrates was observed at about 40 days. There was rise in the RAs after 40 days of the experiment in comparison with that at zero time, while the oxidation efficiency (*k*_cat_) of ABTS (1.9), DAB (2.0) and OPD (2.1) was comparable (*p* < 0.05 in comparison with zero time in all cases). However, after the immunization of mice with MOG for 40 days, there was a strong difference in the increase in the oxidation activity of ABTS (5.6-fold), DAB (2.7-fold), and OPD (1.7 -fold) compared to that at zero time (*p* < 0.05; Figure 3).

Figure 4 demonstrated the changes in the average *k*_cat_ values of the IgGs corresponding to the spontaneous and MOG-induced development of EAE in the 2D2 mice.

As in the case of Th, the best substrate for the IgGs of 2D2 mice is ABTS. At approximately day 40 of the spontaneous development of EAE, the efficiency of the ABTS oxidation by the 2D2 mice IgGs increased by 3.5 times (*p* < 0.05), and after the immunization of the mice with MOG, this was by 6.0 times (*p* < 0.05; Figure 4A). The spontaneous (2.0-fold) and MOG-induced (2.6-fold) development of EAE in the 2D2 mice leads to a smaller increase in the activity of antibodies in the oxidation of DAB (*p* < 0.05; Figure 4B). The initial peroxidase activity of the antibodies in the oxidation of OPD was low; after 40 days of the spontaneous development of EAE, it increased 2.5 times, while after the treatment of mice with MOG, it increased 4.9 times (*p* < 0.05; Figure 4C).

Unfortunately, the amount of IgG preparations for a complete analysis of the peroxide-independent oxidoreductase activity changes during the time of the EAE development and before and after their immunization with MOG was insufficient. According to the data in Figure 3 and Figure 4, the maximum increase in the peroxidase activity occurred approximately at day 40 after the mice were immunized with MOG. Therefore, for an estimation of the H_2_O_2_-independent oxidoreductase activity, we used IgG_mix_ preparations corresponding to zero time (3 months of age) and at 40 days before and after their treatment with MOG (Figure 5).

The oxidoreductase activity of the IgGs mixtures increased after 40 days of spontaneous and MOG-induced EAE and this amount was different for various substrates and lines of mice. To simplify the comparison of the *k*_cat_ values corresponding to different lines of mice and substrates, all of the obtained data are shown in Table 1. This table also includes the data for the C57BL/6 mice that were obtained earlier in [48].

At 3 months of age for the IgGs of the Th (70.0 min^−1^), 2D2 (41.8 min^−1^) and C57BL/6 (69.8 min^−1^) mice, the ABTS turned out to be the better substrate in comparison with DAB (*p* < 0.05): Th (16.4 min^−1^), 2D2 (17.3 min^−1^) and C57BL/6 (5.2 min^−1^) (Table 1). A much lower activity level was observed in the oxidation by the antibodies of the OPD group: Th (0.43 min^−1^) and 2D2 (0.30 min^−1^). At the same time, the increase in activity in the peroxidase oxidation of ABTS 40 days after immunization of mice with MOG in the case of all three lines of mice was comparable, 5.3–6.0-fold (Table 1). After the mice were immunized with MOG, approximately the same increase in the peroxidase-dependent oxidation of DAB was observed for the Th and 2D2 mice (2.4–2.6-fold). Still, in the case of the C57BL/6 mice, it was higher—5.1-fold (Table 1).

Interestingly, the immunization of the Th mice with MOG did not lead to a remarkable increase in the IgG activity in the oxidation of OPD when it was compared with the spontaneous development of EAE: a factor of the increase in the activity in both cases ~2.1 (*p* > 0.05; Table 1). A somewhat different situation was observed for the OPD oxidation by the antibodies from the 2D2 mice; after 40 days of spontaneous EAE development, the efficiency of this substrate oxidation was approximately 2.0 times lower than it was after the immunization of mice (*p* < 0.05) (Table 1). However, in general, all three lines of the mice demonstrate common patterns of an increase in the two redox activities during the spontaneous and MOG-accelerated development of EAE.

## 3. Discussion

It is known that some compounds play an essential role in the processes of oxidative stress [43]. Oxidative disorders are an imprescriptible part of the pathological processes in many autoimmune diseases [49,50], including MS [51,52]. Oxygen species and subsequent oxidative damages could contribute to the development of MS [51]. Reactive oxygen species initiate lesions by inducing the blood–brain barrier destruction, reinforcing myelin phagocytosis, and migrating leukocytes. They contribute to lesion perseverance due to mediating cellular and biological macromolecules damages that are important for the CNS cells’ functioning [49,50,51,52,53,54,55,56].

Endogenous antioxidant enzymes usually counteract oxidative stress. It was shown that in MS that they include overexpressed superoxide dismutase 1 and 2, catalase, and heme oxygenase 1 [57]. There is no doubt that in the case of all mammals, other antioxidant enzymes are also directed against oxidative stress. However, antioxidant enzymes are mostly present in different cells, and their several enzymatic activity levels (catalase, peroxidase, glutathione peroxidases, and other enzymes oxidizing different compounds) in the blood are low because, in the blood, they quickly lose their activities [28,29]. Stable blood IgGs are present in the blood in high concentrations (5.4–18.2 g/L) and remain intact for several months [30]. It was previously shown that the blood of healthy donors and rats contains antibodies-abzymes with antioxidant peroxidase and oxidoreductase activities [31,32,33,34,35,36,37,38,39,40]. In addition, it was demonstrated that abzymes with these redox activities in the blood of MS and SLE patients are statistically higher than they are in healthy donors [39]. However, for the analysis of these activities, the IgGs were taken from patients’ sera of blood at different stages of their AIDs development. Therefore, it was interesting to analyze how antibodies with peroxidase and oxidoreductase activities can be changed during different stages of AIDs development. For this purpose, we first used an analysis of the changes that occur over time in the C57BL/6 mice antibodies peroxidase activity during the spontaneous and antigen-induced development of EAE [48]. Here, we compared the data for the C57BL/6, Th, and 2D2 mice.

It was shown previously that the peroxidase activity of IgGs from the blood of healthy donors and patients with SLE and MS [37,38,39] and C57BL/6 [48] mice is remarkably or significantly higher than the oxidoreductase activity is of the same polyclonal Abs. Similar results were obtained in the case of the Th and 2D2 mice (Table 1). Interestingly, the average relative activity of the IgG preparations from healthy humans in the oxidation of ABTS (73.7 ± 12.0 min^−1^ [37]) is comparable with that for the three-month-old C57BL/6 (69.8 ± 23.0 min^−1^ [48]), Th (70.0 ± 21.0 min^−1^) and 2D2 (41.8 ± 2.5 min^−1^) mice before their immunization (Table 1). In SLE and MS patients, the relative activity of IgGs in ABTS oxidation is about 4.5 times higher than it is in healthy donors [39]. The immunization of mice with MOG leads to the development of EAE and an increase/a comparable rise in the IgG activity (-fold): C57BL/6 (5.3;), Th (5.6), and 2D2 (6.0) (Table 1).

The spontaneous and MOG-accelerated development of EAE which is associated with the production of various abzymes begins after the changes in the differentiation profile of the brain stem cells of mice. Interestingly, the profiles of stem cell differentiation in the case of the C57BL/6, Th, and 2D2 mice are markedly or even very different (Appendix A). Nevertheless, the spontaneous and MOG-induced development of them ultimately leads to an EAE pathology, which is characterized by similar or nearly the same immunological and medical parameters [9,10,11]. The diversity in the bone marrow stem cells differentiation profiles during the spontaneous and MOG-stimulated development of EAE leads to the production of abzymes hydrolyzing DNA, MOG, and MBP [9,10,11]. The immunization of the C57BL/6 and Th mice with MOG leads to a substantial increase in the activity of the antibodies in the hydrolysis of DNA, MOG, and MBP (Appendix A). At the same time, the treatment of the 2D2 mice with MOG leads to a strong suppression of the activity of the abzymes in the hydrolysis of DNA, MOG, and MBP when it is compared with the spontaneous development of EAE (Appendix A).

Interestingly, there are significant differences in the patterns of changes in the relative activity of the abzymes in the hydrolysis of DNA, MBP, and MOG on the one hand, and redox activity, on the other. The increase in the production of abzymes hydrolyzing DNA, MBP, and MOG after the immunization of the C57BL/6 mice with MOG occurred mainly from 7 to 20 days, corresponding to the onset and acute phase of EAE (Appendix A). During the spontaneous development of EAE and after the C57BL/6 mice were immunized with MOG, a sharp increase in the peroxidase activity during the onset and acute phase of the pathology was not revealed; in both cases, a gradual increase in peroxidase activity was observed (Appendix A).

After the immunization with MOG, the Th mice showed a sharp increase in the activity of the IgGs in the hydrolysis of DNA, MBP and MOG from about 7 to 30 days (Appendix A) [10]. During the spontaneous development of EAE, the efficiency of ABTS oxidation increased smoothly. At the same time, treating the mice with MOG leads to a very strong increase in this activity for up to 40 days (Figure 3A).

During the spontaneous development of EAE, in general, an increase in the activity of the antibodies in the hydrolysis of DNA, MBP and MOG was observed (Appendix A). A completely different situation was observed in the case of the 2D2 mice. Unlike the C57BL/6 and Th immunization of which resulted in a strong increase in the hydrolyzing activities (Appendix A), the treatment of the 2D2 mice with MOG led to a sharp decrease in the activity of the antibodies in the hydrolysis of DNA, MBP and MOG (Appendix A). At the same time, as in the case of the C57BL/6 and Th mice, the peroxidase activity of the 2D2 mice antibodies increased very strongly for up to 40–50 days (Figure 4). Thus, after the immunization of the mice with MOG, there was a similarity for the C57BL/6, Th, and 2D2 mice only in the case of the increase in the peroxidase activity.

The origin of abzymes with redox activities is not clear yet. It is possible to propose several ways in which they originate, theoretically. First, Abs-abzymes with various redox activities in healthy humans and patients with AIDs can simply reflect the constitutive production of germline antibodies as described by the Paul group [58,59]. In addition, the synthesis of such abzymes can probably be stimulated by various mutagenic, carcinogenic, and other toxic compounds in their complexes with some proteins, which can lead to the synthesis of abzymes against these haptens. In addition, the anti-idiotypic second antibodies against different enzymes’ catalytic sites may also possess catalytic activities [12,13,14,15,16,17]. Therefore, one cannot exclude that different abzymes with redox activities could be anti-idiotypic Abs against canonical peroxidases, catalases, superoxide dismutases, glutathione peroxidases, and other enzymes oxidizing different compounds.

It should be noted that healthy people’s and animals’ blood, they usually contain no abzymes with DNase and high protease or amylase activities from healthy donors Protease activities including the hydrolysis of vasoactive neuropeptide and amylase activities abzymes from healthy donors, which are significantly lower than they are in patients with autoimmune diseases [14,15,16,17]. Antibodies catalyzing redox reactions demonstrate high activity even in healthy donors [32,33,34,35,36,37,38,39,40].

The appearance of abzymes with different high enzymatic activities (DNase, RNase, protease, amylase, and other) [12,13,14,15,16,17], as well as an increase in the activity of Abs with redox activities, is clearly associated with the development of various AIDs.

The imperfection of the antioxidant system can lead to abnormalities occurring in the hypofunction of the NMDA receptors and myelination, the death of interneurons, and various other abnormal processes, and thus, lead to the development of multiple pathologies [51,55,56]. The Abs of healthy donors and animals, as well as patients with AIDs, possess not only peroxidase but also catalase, superoxide, dismutase, and H_2_O_2_-independent oxidoreductase activities [12,13,14,15,16,17,31,32,33,34,35,36,37,38,39,40]. Abzymes with redox activities oxidizing different compounds that are harmful to humans could reduce the oxidative pathological processes due to them protecting the body from oxidative stress. Due to the high concentration of them in blood and the prolonged circulation of antibodies with redox activities, they can decrease the oxidative disorders in the bloodstream. In addition, the ability of the Abs to be accumulated in the foci of inflammation, catalytic IgGs with peroxidase and catalase activities, together with canonical catalases and glutathione peroxidases [60] can enable them to participate in the regulation of hydrogen peroxide concentration and limit the damage that is caused by a high concentration of H_2_O_2_ oxidative processes in these zones. Therefore, the question arises: why are abzymes with redox functions needed?

The biological role of abzymes with redox activities in humans and animals may be much more essential than it might seem to be at first glance. It has been shown that Abs of different mammals possess a superoxide dismutase activity and efficiently reduce the amount of singlet oxygen (^1^O_2_*) to ^•^O_2_ leading to the formation of H_2_O_3_ as the first reaction intermediate, while a unique cascade of chemical reactions occurs, thus finally resulting in the formation of H_2_O_2_ [61,62]. It was proposed that the origin of this redox enzymatic activity is based on the adaptive maturation of the variable domains of antibodies. These data demonstrate a possible protective function of such abzymes and raise the question of whether the detoxification of ^1^O_2_* could play an important role in the evolution of Abs. These abzymes demonstrate the additional mechanisms of how oxygen can be reduced and recycled in the phagocyte action, thus increasing the immune system’s anti-microbicidal action [61,62]. It seems likely that in healthy mammals, there exists a whole set of Abs-abzymes with various redox protective functions. Some abzymes with superoxide dismutase activity can convert oxygen into H_2_O_2_, while other Abzs with peroxidase and catalase activities neutralize hydrogen peroxide. Taking into account a set of various compounds that are dangerous to mammals which may be substrates of redox IgGs [31,32,33,34,35,36,37,38,39,40], it is easy to suggest that the set of possible substrates of abzymes with antioxidant functions can be extremely wide. Taken together, natural abzymes can serve as an additional system of detoxification of the reactive oxygen species in mammals.

## 4. Materials and Methods

### 4.1. Reagents

The chemicals, Protein G-Sepharose (17061801), Protein A-Sepharose (17046901), and the Superdex 200 HR 10/30 (17-5175-01) columns were purchased from GE Healthcare Life Sciences (New York, NY, USA). DAB, 3,3′-diaminobenzidine (Sigma-Aldrich, St. Louis, MO, USA, D8001); ABTS, 2,2′-azino-bis(3-ethylbenzothiazoline-6-sulfonic acid) diammonium salt was purchased from Sigma-Aldrich (St. Louis, MO, USA, A1888); we used OPD, o-phenylenediamine also (Sigma-Aldrich, P9029). These preparations were free from nucleic acid, lipids, oligosaccharides, and other possible contaminants. MOG_35–55_ (2568/1) was from EZBiolab (Munich, Germany), while the *Bordetella pertussis* toxin (*M. tuberculosis;* BML-g100-0050) was from Native Antigen Company (Oxfordshire, UK).

### 4.2. Experimental Animals

The C57BL/6, Th, and 2D2 mice (3 months of age; in each group, there were 10 mice) were grown in a special mouse breeding facility of the Institute of Cytology and Genetics (ICG) using standard conditions that were free of any bacterial, viral, and other pathogens. The Bioethical Institute of Cytology and Genetics committee supported our study. We carried out all of the experiments following the ICG Bioethical committee’s protocols which are adequate to the humane principles of different experiments with animals of the European Community Council Directive 86/609/CEE.

### 4.3. Immunization of Mice

The IgGs were analyzed by us, and they were earlier used for the analysis of various parameters, showing the development of EAE in C57BL/6 [6,7,8], Th [9,10], and 2D2 [11] mice before and after their treatment with MOG. On the first day (zero time—zero day, everywhere, this is the basal day of induction; all of the mice 3 months of age were used in each analyzed group), the mice that were used were immunized with MOG, as described earlier [6,7,8,9,10]. The control mice at 3 months of age were not immunized; these mice were used to analyze the parameters during the spontaneous development of EAE. For the purification of the IgGs, 0.6–0.9 mL of the blood was collected 7–90 days after their treatment by the standard decapitation of mice [6,7,8,9,10,11]. The methods for the immunization of the mice are described in [6,7,8,9,10], and their detailed description is given in the Appendix A (Part 1, Immunization of mice).

### 4.4. IgG Purification

Electrophoretically and immunologically homogeneous IgGs were obtained earlier in [6,7,8,9,10,11] by the chromatography of mice blood plasma proteins first on Protein G-Sepharose and then, additionally by FPLC gel filtration in severe conditions (pH 2.6), thus providing the dissociation of the protein complexes. For the protection of the IgG preparations from potential bacterial and viral contamination, the Abs were filtered using 0.1 μm Amicon filters. The solutions that were obtained were divided into many aliquots and kept at −70 °C in sterilized tubes before the IgGs were used in various experiments. An SDS-PAGE analysis of the IgG samples for homogeneity was performed using 4–15% gradient gels; all of the proteins were visualized by silver staining [6,7,8,9,10,11]. The IgG preparations were separated by SDS-PAGE to exclude potential artefacts due to possible traces of canonical enzymes. Their protease, nuclease, and redox activities were detected using a gel assay, as in [6,7,8,9,10,11]. These activities were detected only in the band of the intact IgGs; no other peaks of proteins, redox, protease, or DNase activities were revealed.

The methods for conducting IgG purifications have been published earlier [6,7,8,9,10,11], and their more detailed description is given in the Appendix A (Part II, IgG purification).

### 4.5. Assay of Redox Activities

Estimating the relative peroxide-independent oxidoreductase and H_2_O_2_-dependent peroxidase activities of the IgGs was performed under optimal conditions [31,32,33,34,35]. The reaction mixtures (95–150 µL) contain 25 mM K-phosphate buffer (pH 6.8), with or without 10 mM H_2_O_2_, one of three different substrates (0.07–0.55 mM) and 0.05–0.4 mg/mL of IgGs. The oxidation of the ABTS (diammonium salt of 2,2′-azino-bis(3-ethylbenzothiazoline-6-sulfonic acid), DAB (3,3′-diaminobenzidine), and OPD (o-phenylenediamine) were detected by the changes in optical density at 450 nM (A_450_) using quartz cuvettes (0.1 cm) and a Genesis Bio spectrophotometer 10S (Thermo Fisher Scientific, Inc.; Pittsburgh, PA, USA). All of the mixtures were incubated at 23 °C; the time dependencies of the changes in A_450_ (0.5–25 min) were analyzed. As controls, the reaction mixtures without IgGs were used. Errors in the values were within 5–9%. The apparent values of the catalytic constants (*k*_cat_) were estimated using the maximum possible concentrations of the substrates used, at which no precipitation of the hydrolysis products occurred. The *k*_cat_ values were calculated according to the equation: *k*_cat_ = V (M/min)/ [IgG] (M).

### 4.6. In Situ Analysis of Peroxidase Activity

To exclude possible traces of contaminating canonical redox enzymes, the IgGs were separated by SDS-PAGE. Their redox activities were revealed using a gel assay, according to [6,7,8,9,10,11]. The SDS-PAGE analysis of the IgGs (16 μg) homogeneity and in situ redox activity was performed in 3–16% gradient gels (0.12% SDS) under non-reducing conditions in the absence of DTT [31,32,37,38,39]. To restore the redox activities, the SDS was removed after the SDS-PAGE by gel incubation for 1.0 h at 23 °C with K-phosphate buffer (pH 6.8). The gel was washed nine times with this buffer. To allow for the Abs refolding and to assay for the redox activities, several gel longitudinal slices were incubated in the standard reaction mixture containing 0.2 mg/mL DAB and 10 mM H_2_O_2_ for 10 h until a coloured insoluble product of DAB oxidation was formed. The parallel longitudinal gel lanes were used to detect the IgG position on the gel by silver staining.

### 4.7. Statistical Analysis

The results are given as the mean and standard deviation of at least 2–3 independent experiments for each IgG preparation, averaged over 10 different mice in every group. The differences between the characteristics of the IgG samples of various groups were estimated by the non-parametric Kruskal–Wallis one-way analysis of variance; *p* < 0.05 was considered statistically significant. The median (M) and interquartile ranges (IQR) were estimated.

## 5. Conclusions

Taken together, autoimmune-prone C57BL/6, Th, and 2D2 mice are characterized by the spontaneous and MOG-accelerated EAE development that is associated with changes in the differentiation of bone marrow stem cells and the production of abzymes hydrolyzing MBP, MOG, and DNA. The spontaneous development of EAE leads to an increase in the activity of antibodies-abzymes, with peroxidase and oxidoreductase activities oxidizing different substrates of canonical redox enzymes. The immunization of these mice with MOG leads not only to the acceleration of EAE development, but also to a strong increase in the redox activities of the antibodies. It was shown that in patients with autoimmune diseases, the redox activity of the antibodies is also higher than it is in healthy donors. Thus, in MS, SLE patients, and the EAE-prone mice, the abzymes with an increased redox activity can protect them from some harmful oxidants somewhat better than they can in healthy people and animals.

## Figures and Tables

**Figure 1 molecules-27-07527-f001:**
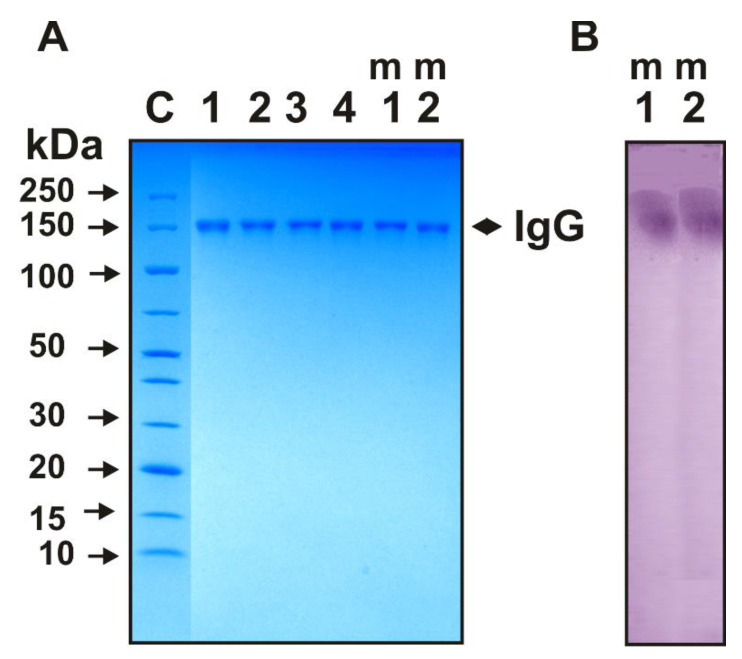
SDS-PAGE analysis of 14 µg IgGs electrophoretic homogeneity of four individual Abs (lanes 1–4) and equimolar mixtures of 30 preparations corresponding IgGs from Th (lane m1; Th-IgG_mix_) and 2D2 (lane m2; 2D2-IgG_mix_) mice using nonreducing 4–15% gradient gel (**A**). Lane C corresponds to a mixture of proteins with known molecular weights (**A**). In situ peroxidase activity analysis of m1 and m2 IgG_mix_ (**B**). After SDS-PAGE, two longitudinal strips of gel were incubated using special conditions for protein refolding. Then, the peroxidase activity was revealed by incubating gel slices for 10 h in the standard reaction mixture containing DAB and H_2_O_2_ until a coloured insoluble oxidation product was formed.

**Figure 2 molecules-27-07527-f002:**
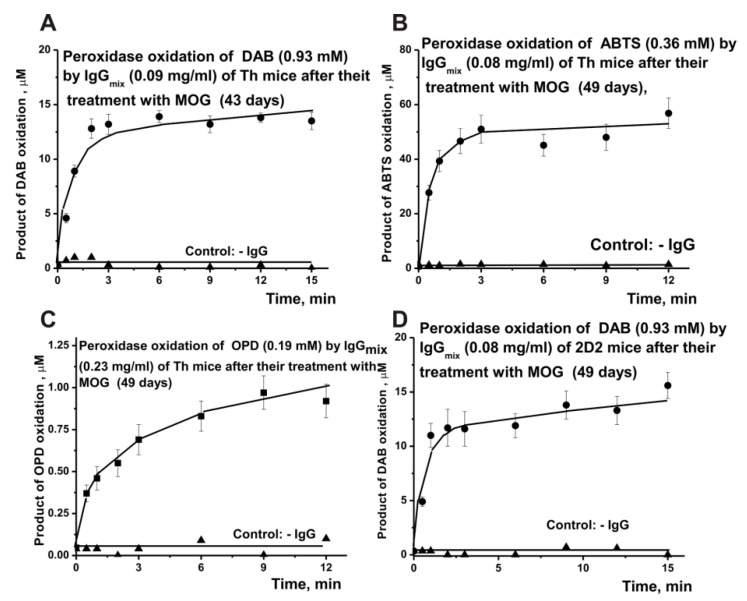
Typical examples of the time dependences of different product formations in the reaction of three substrates (ABTS, DAB, and OPD) due to their oxidation by IgG_mix_ preparations in the presence of H_2_O_2_. All of the designations are given in Panels (**A**–**D**).

**Figure 3 molecules-27-07527-f003:**
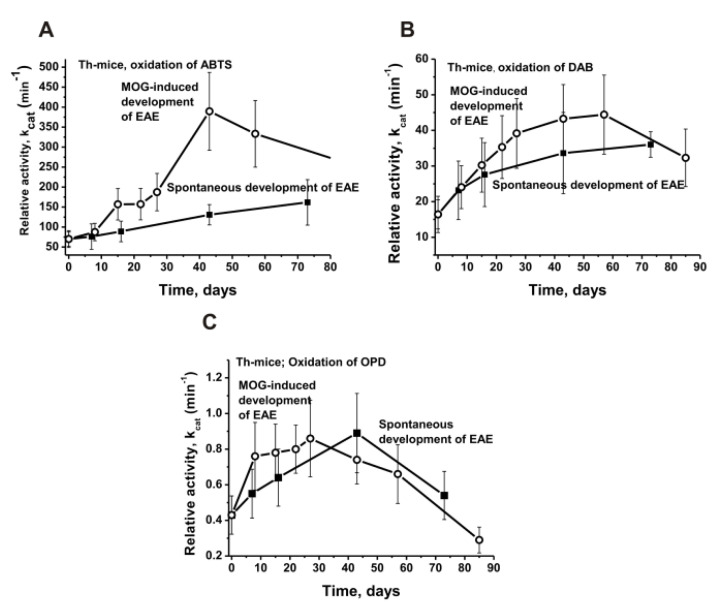
Average changes over time in the relative activity in the oxidation of ABTS (**A**), DAB (**B**), and OPD (**C**) before and after Th mice immunization with MOG. Spontaneous development of EAE corresponds to untreated mice. All of the designations are marked in three panels. The mean values of *k*_cat_ corresponding to 10 mice of each group are presented.

**Figure 4 molecules-27-07527-f004:**
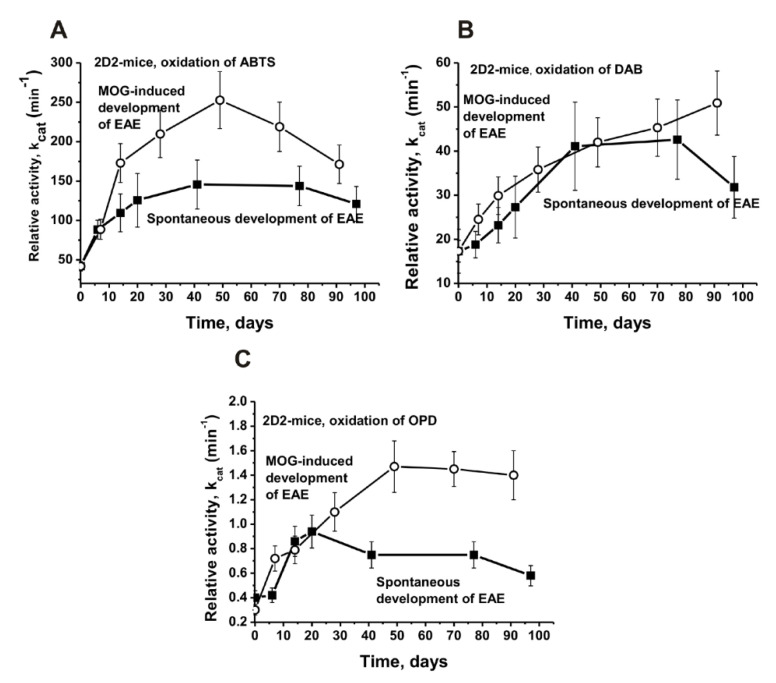
Average changes over time in the relative activity in the oxidation of ABTS (**A**), DAB (**B**), and OPD (**C**) before and after 2D2 mice immunization with MOG. All of the designations are marked in three panels. The mean values of *k*_cat_ corresponding to 10 mice of each group are presented.

**Figure 5 molecules-27-07527-f005:**
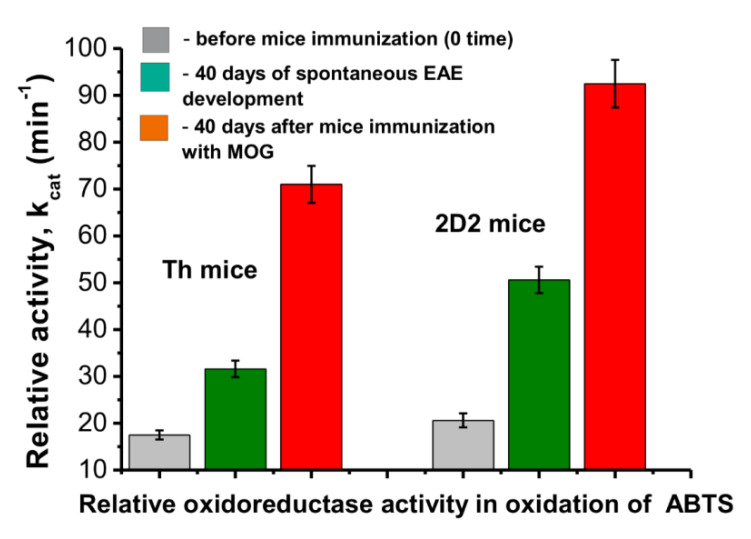
Relative values of *k*_cat_ characterizing the rate of ABTS oxidation by IgGs from the blood of Th and 2D2 mice in the absence of hydrogen peroxide (oxidoreductase activity) at the zero time of the experiment and 40 days after spontaneous development of EAE and immunization of mice with MOG. To estimate the *k*_cat_ values, equimolar mixtures of 10 IgG preparations corresponding to each of the groups were used.

**Table 1 molecules-27-07527-t001:** Relative peroxidase and oxidoreductase activities IgGs of Th, 2D2, and C57BL/6 mice.

Substrate	Values of *k*_cat_, min^−1^					
	Zero Time (3 Months of Age)	40 Days after Spontaneous EAE Development	40 Days after Treatment with MOG	Ratio of 2 to 1	Ratio of 3 to 1	Ratio of 3 to 2
	1	2	3			
	**H_2_O_2_-dependent peroxidase activity ***					
	**Th mice**					
ABTS	70.0 ± 21.0	130.8 ± 25.4	389.5 ± 97.3	1.9	5.6	3.0
DAB	16.4 ± 5.1	33.6 ± 11.4	43.3 ± 9.6	2.0	2.6	1.3
OPD	0.43 ± 0.1	0.89 ± 0.2	0.89 ± 0.1	2.1	2.1	1.0
	**2D2 mice**					
ABTS	41.8 ± 2.5	145.6 ± 2.5	252.7 ± 36.1	3.5	6.0	1.7
DAB	17.3 ± 5.0	41.1 ± 10.0	42.0 ± 5.6	2.4	2.4	1.0
OPD	0.30 ± 0.06	0.75 ± 0.1	1.5 ± 0.2	2.5	5.0	2.0
	**C57BL/6 mice** [48]					
ABTS	69.8 ± 23.1	100.4 ± 12.0	372.0 ± 40.0	1.4	5.3	3.7
DAB	5.2 ± 1.0	12.3 ± 1.2.	26.7 ± 7.0	2.4	5.1	2.2
	**H_2_O_2_-independent oxidoreductase activity ****					
	**Th mice**					
ABTS	17.5 ± 1.0	31.6 ± 1.8	71.0 ± 1.9	1.8	4.1	2.2
	**2D2 mice**					
ABTS	20.6 ± 1.5	50.6 ± 2.8	92.5 ± 5.1	2.5	4.5	1.8

* Mean values estimated using data from 10 mice; ** Values estimated using data for IgG_mix_ (mixtures of IgGs from 10 mice).

## Data Availability

The data supporting our study results are included in the article and Appendix A containing Appendix A.

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
