# Peer review of "Antibodies-Abzymes with Antioxidant Activities in Two Th and 2D2 Experimental Autoimmune Encephalomyelitis Mice during the Development of EAE Pathology"

_molecules, 2022, doi:10.3390/molecules27217527_

Round 1
Reviewer 1 Report
1. The title of the manuscript is too long. It should be revised and make it more concise.
2. The context of the abstract is not in accordance with the title and purpose of the manuscript. It should be revised.
3. In introduction section, there are several sentences that are not understandable. For example, “The parcel reduced oxygen species in various organisms…….”
4. It is important to write the catalogue numbers of assay kits along with the names of manufacturers.
5. The text written in figure 2 is not clear. Please increase the font size of the text in figure.
6. What is the difference between the peroxidase activity and enzymatic activity?
7. In figure 3, the values of errors bars are beyond the maximum value of y-axis bar. Please reconsider the maximum value of y-axis in this figure.
8. In the results section, the authors have used the term fold-increase. What is the difference between the fold-increase and percent-increase?
9. It is suggested that the authors should use the single value to differentiate the statistical results.
10. The authors have obtained the good results but failed to interpret them in the results section. Similarly, the discussion is not in accordance with the key findings their study. In the discission section, usually, the key findings of own experiments are usually discussed in the light of already known in that particular area. I suggest the authors to reconsider these sections carefully.
11. Similarly, the conclusion of this study in its present form does not reflect the key findings of the study. It should reflect the overall key findings in accordance with future perspectives and its impact on the society.
12. What are the limitations of your study?
13. There are several grammatical mistakes and syntax errors. The whole manuscript needs critical revision to remove all the grammatical mistakes and syntax errors.
Author Response
1
- The title of the manuscript is too long. It should be revised and make it more concise.
Answer: It was done
- The context of the abstract is not in accordance with the title and purpose of the manuscript. It should be revised.
Answer: It was done
- In introduction section, there are several sentences that are not understandable. For example, “The parcel reduced oxygen species in various organisms…….”
Answer: It was done
- Several different oxygen species in various organisms (O2·-, H2O2, and OH·) are intermediates of aerobic respiration
- It is important to write the catalogue numbers of assay kits along with the names of manufacturers.
Answer: It was done
- The text written in figure 2 is not clear. Please increase the font size of the text in figure.
Answer: It was done
- What is the difference between the peroxidase activity and enzymatic activity?
Answer: All enzyme activities can be called enzymatic activities. But different parts of the text mean different activities. Taking this into account, in parentheses after the enzymatic activity, data are added on what exactly
- In figure 3, the values of errors bars are beyond the maximum value of y-axis bar. Please reconsider the maximum value of y-axis in this figure.
Answer: It was done
- In the results section, the authors have used the term fold-increase. What is the difference between the fold-increase and percent-increase?
Answer: You can specify from how many to how many kcat were increased, but more clearly and understandably how many times (fold) the kcat values increased. But all kcat values are given in Table 1 and the coefficient of increase in values
- It is suggested that the authors should use the single value to differentiate the statistical results.
Answer: Sorry, but we did not understand this remark, the same P < 0.05value is given everywhereP < 0.05
- The authors have obtained the good results but failed to interpret them in the results section. Similarly, the discussion is not in accordance with the key findings their study. In the discission section, usually, the key findings of own experiments are usually discussed in the light of already known in that particular area. I suggest the authors to reconsider these sections carefully.
Answer: Sorry, but in terms of redox activities during the development of autoimmune diseases, this is only the second article. A more detailed discussion with what literature data, which are not yet available, at this stage seems to be very difficult and, from our point of view, premature. There is also no indication in your remark in what direction the obtained data could be discussed.
- Similarly, the conclusion of this study in its present form does not reflect the key findings of the study. It should reflect the overall key findings in accordance with future perspectives and its impact on the society.
Answer: At this stage, this is also very difficult to do. For now, we can only say that in patients with autoimmune diseases, the redox activity of antibodies is also higher than in healthy donors. Thus, in MS, SLE patients, and EAE-prone mice, abzymes with increased redox activity can protect them from some harmful oxidants somewhat better than in healthy people and animals.
13.What are the limitations of your study?
Answer: mouse is only a model to explore the possibility of studying the increase in the activity of abzymes in the oxidation of causal substrates. In people with atsutoimmune diseases, the activity of such abzymes is also increased. But so far we have difficulties in analyzing activities with redox activities in people at different stages of development of autoimmune diseases.
- There are several grammatical mistakes and syntax errors. The whole manuscript needs critical revision to remove all the grammatical mistakes and syntax errors.
Answer: We tried to correct all the shortcomings in this regard.
Thanks a lot for the very helpful comments.
Professor George A. Nevinsky
Reviewer 2 Report
The article by Anna S. Tolmacheva et. al. entitled " Increase in antibodies-abzymes with antioxidant eroxidase and oxidoreductase activities in two Th and 2D2 experimental 3 autoimmune encephalomyelitis mice during the development 4 of EAE pathology " is an interesting study on discovery that catalytic antibodies having antioxidant peroxidase and oxidoreductase activities can work as antioxidant peroxidase and oxidoreductase activities. The authors analyzed changes in these redox activities in blood IgG of Th and 2D2 mice corresponding to different stages of EAE development. The author found that the peroxidase activity is higher than the oxidoreductase activity during developing EAE. Furthermore, Furthermore, they found that the peroxidase activity of IgG in MOG-immunized mice was about 6-fold higher than before immunization. They argued that IgG peroxidase and oxidoreductase activity in EAE mice could play an important role in protection from toxic compounds and oxidative stress. In general, the article is written clearly and presents interesting data, even if the figure is not well represented. This article would be of interest to scientists who are focusing on the study of catalytic antibody and autoimmune disease.
I have the following comments and concerns.
1. In page 11 line 399, what caused the sharp decrease in hydrolytic activity of these antibodies?
2. In page 11 line 415, “It should be noted that healthy people's and animals' blood usually contains no any abzymes except catalyzing redox reactions.” What does this mean? Catalytic antibodies with peptidase activities have been found in the blood of healthy individuals.
3. The size of Figure 2 in page 6 is very small and very difficult to read. Please present appropriate size.
I am not able to find supplement figures (Figure S1-S7). Please provide them. I found several inconsistent abbreviations used, eg. autoimmune disease (AD) or autoimmune (AI) disease. There are many careless grammatical and typographical errors throughout the manuscript. I would encourage the authors to proofread the manuscript carefully.
Author Response
The article by Anna S. Tolmacheva et. al. entitled " Increase in antibodies-abzymes with antioxidant eroxidase and oxidoreductase activities in two Th and 2D2 experimental 3 autoimmune encephalomyelitis mice during the development 4 of EAE pathology " is an interesting study on discovery that catalytic antibodies having antioxidant peroxidase and oxidoreductase activities can work as antioxidant peroxidase and oxidoreductase activities. The authors analyzed changes in these redox activities in blood IgG of Th and 2D2 mice corresponding to different stages of EAE development. The author found that the peroxidase activity is higher than the oxidoreductase activity during developing EAE. Furthermore, Furthermore, they found that the peroxidase activity of IgG in MOG-immunized mice was about 6-fold higher than before immunization. They argued that IgG peroxidase and oxidoreductase activity in EAE mice could play an important role in protection from toxic compounds and oxidative stress. In general, the article is written clearly and presents interesting data, even if the figure is not well represented. This article would be of interest to scientists who are focusing on the study of catalytic antibody and autoimmune disease.
I have the following comments and concerns.
- In page 11 line 399, what caused the sharp decrease in hydrolytic activity of these antibodies?
Answer: This question is very difficult to answer at present, because it is not yet clear how different changes in the differentiation profile of bone marrow stem cells are reflected in different mice on the production of antibodies to different antigens.
- In page 11 line 415, “It should be noted that healthy people's and animals' blood usually contains no any abzymes except catalyzing redox reactions.” What does this mean? Catalytic antibodies with peptidase activities have been found in the blood of healthy individuals.
Answer: it was corrected
- The size of Figure 2 in page 6 is very small and very difficult to read. Please present appropriate size.
Answer: it was corrected
- I am not able to find supplement figures (Figure S1-S7). Please provide them. I found several inconsistent abbreviations used, eg. autoimmune disease (AD) or autoimmune (AI) disease. There are many careless grammatical and typographical errors throughout the manuscript. I would encourage the authors to proofread the manuscript carefully.
ADs changed for AIDs, while AI is removed.
Answer: Figures S1-S7 were added in Supplementary materials
Thanks a lot for the very helpful comments.
Professor George A. Nevinsky
Round 2
Reviewer 1 Report
The authors have revised their manuscript in accordance with the comments raised in previous revision. There is no further comments comments from my side.